# The Role of Climate Niche, Geofloristic History, Habitat Preference, and Allometry on Wood Density within a California Plant Community

**Rebecca A. Nelson [1,\*], Emily J. Francis [2], Joseph A. Berry [3], William K. Cornwell [4] and Leander D. L. Anderegg [3,5]**

[1] Department of Biology, School of Humanities and Sciences, Stanford University, Stanford, CA 94305, USA

[2] Department of Integrative Biology, College of Natural Sciences, University of Texas at Austin, Austin, TX 78712, USA; emily.francis@austin.utexas.edu

[3] Department of Global Ecology, Carnegie Institution for Science, Stanford, CA 94305, USA; jberry@carnegiescience.edu (J.A.B.); leanderegg@gmail.com (L.D.L.A.)

[4] School of Biological, Earth and Environmental Sciences, University of New South Wales, Sydney, NSW 2052, Australia; w.cornwell@unsw.edu.au

[5] Department of Integrative Biology, College of Letters & Science, University of California, Berkeley, CA 94720, USA

\* Correspondence: rnelson3@stanford.edu

**Abstract:** *Research Highlights:* To better understand within-community variation in wood density, our study demonstrated that a more nuanced approach is required beyond the climate–wood density correlations used in global analyses. *Background and Objectives:* Global meta-analyses have shown higher wood density is associated with higher temperatures and lower rainfall, while site-specific studies have explained variation in wood density with structural constraints and allometry. On a regional scale, uncertainty exists as to what extent climate and structural demands explain patterns in wood density. We explored the role of species climate niche, geofloristic history, habitat specialization, and allometry on wood density variation within a California forest/chaparral community. *Materials and Methods:* We collected data on species wood density, climate niche, geofloristic history, and riparian habitat specialization for 20 species of trees and shrubs in a California forest. *Results:* We found a negative relationship between wood density and basal diameter to height ratio for riparian species and no relationship for non-riparian species. In contrast to previous studies, we found that climate signals had weak relationships with wood density, except for a positive relationship between wood density and the dryness of a species' wet range edge (species with drier wet range margins have higher wood density). Wood density, however, did not correlate with the aridity of species' dry range margins. Geofloristic history had no direct effect on wood density or climate niche for modern California plant communities. *Conclusions:* Within a California plant community, allometry influences wood density for riparian specialists, but non-riparian plants are 'overbuilt' such that wood density is not related to canopy structure. Meanwhile, the relationship of wood density to species' aridity niches challenges our classic assumptions about the adaptive significance of high wood density as a drought tolerance trait.

**Keywords:** wood density; allometry; functional traits; climate niches

---

## 1. Introduction

Wood density (the mass per unit volume of wood) is considered an essential functional trait associated with mechanical support, carbon and nutrient storage, drought tolerance, water transport,

and pathogen defense [1–3]. Variation in wood functional traits often reflect ecological tradeoffs that are influenced by allometric, biogeographic, and phylogenetic factors [1]. Understanding wood density patterns can help elucidate spatial trends in aboveground biomass [4]. Moreover, wood density is a useful predictor of growth and mortality rates in the tropics [5] and is related to growth and performance in Mediterranean climates [6]. Finally, models of carbon stocks often rely on wood density, making estimates of wood density operationally as well as ecologically important [7,8].

Wood density often relates to mechanical stability [9–11] with crown architecture, diameter, and height correlating with wood density at small scales [8,12]. Tree height is a key trait for understanding patterns in wood density and allometry in Mediterranean communities [6]. Allometric equations that relate wood density to height and diameter have explained site-specific patterns across tropical ecosystems [8,13]. The relationship of wood density to diameter–height allometry at an individual site level has varied across studies. Whereas some studies have found that tree height at a given diameter is unrelated to wood density [14,15], Iida et al. (2012) found that, at a constant tree height, higher wood density is associated with smaller tree diameters [16], and wood density is positively related to diameter-corrected tree height in rainforests [17]. Simultaneously, in a California mixed evergreen forest, wood density correlated strongly with vessel traits and microsite soil moisture [18,19], indicating that moisture availability can be an environmental filter for wood density at very small scales. The variation in the wood density–allometry relationship across different climate regimes and site conditions may be partially explained by the availability of water at particular microsites, with some individuals and species at or close to drainages being able to access consistent water throughout the year, while other species experience strong seasonal fluctuations in water availability. In other words, riparian habitat specialization could influence the degree to which structural constraints and climate niche relate to wood density.

Global and regional meta-analyses suggest that higher wood density correlates with lower precipitation and higher mean annual temperature [6,11,20]. Across a latitudinal gradient in Central America, patterns in wood density more strongly correlated to precipitation and aridity than temperature [21]. Wood density is positively associated with aridity in New Caledonia, and a global meta-analysis [22,23] suggests that high wood density is adaptive in dry environments. However, precipitation and temperature explain variation in tree height–diameter relationships across North America as well, possibly influencing wood density indirectly through structural constraints [24]. The findings that rainfall amount can modify wood density–allometry relationships at a regional level in water-stressed climates suggests at least some direct effect of climate on wood density [25], and denser wood is correlated with lower drought mortality [26]. Variations in shade tolerance and altitude add further nuance to relationships between wood density and rainfall patterns in tropical and Mediterranean climates [12,27].

While many studies have examined the relationship between climate and wood density across a broad geographic scale, and a number of tropical studies have highlighted the importance of structural or allometric constraints on wood density at a site level, few studies have assessed both structural and climatic constraints on wood density in temperate communities. Moreover, phylogenetic signal, biogeography, and geofloristic history (plant assemblages recorded in the fossil record) can strongly structure the composition and functional traits of temperate plant communities [11,22,27,28]. As a result, community composition that is the contingent outcome of past climates could drive functional trait patterns that are not mechanistically linked to the contemporary climate [28]. In water-stressed ecosystems, uncertainty exists as to whether mechanical demands, habitat specialization, geofloristic history, or macroclimatic niche best explain variation in wood density.

The California floristic province, with its high plant diversity and endemism, is vulnerable to increasing drought frequency and intensity exacerbated by anthropogenic climate change. Microsite variation in soil moisture influences the wood density and tree height of the local community across microsites in northern California [18,19]. This suggests environmental filtering for higher wood density species in drier sites. The current species composition of California mixed evergreen forests

resulted from the confluence of several geofloras (plant community assemblages identified in the fossil record), especially the temperate Arcto-Tertiary geoflora and the subtropical Madro-Tertiary geoflora [29,30]. The Madro-Tertiary and Arcto-Tertiary geofloras, both originating in the Tertiary Period, are distinguished by their biogeographic origin and differing community compositions. The Madro-Tertiary geoflora, consisting mostly of subtropical, sclerophyll species, is associated with the development of a Mediterranean climate in California and is the ancestor of many extant chaparral specialists [29,30]. The Arcto-Tertiary geoflora, compromised of mixed deciduous forest species, advanced into California from more northerly climates [29,30].

We investigated whether structure, climatic niche, geofloristic history, and habitat specialization related to wood density variation among members of a regional woody plant community, sampling 20 species from a northern California plant community across multiple sites. We tested four hypotheses: (1) We hypothesize that the structural demand of supporting a plant canopy will constrain wood density values such that species with tall, thin stems must have higher wood density to support their canopies. We predict that 'taper' (the ratio of plant basal diameter to plant height) will be negatively related to wood density. (2) We predict that, within a community, more arid adapted species will have higher wood density. Thus, we predict that wood density and metrics of water availability will be negatively related for the dry and wet edges of species' aridity niche, with stronger relationships at the dry edge. (3) We hypothesize that species with a Madro-Tertiary geofloristic history would have stronger wood density–climate relationships than species with an Arcto-Tertiary geofloristic history because the Madro-Tertiary geoflora arose in conjunction with California's dry Mediterranean climate. (4) Finally, we hypothesize that riparian habitat specialization may greatly influence the relationship between wood density and structure as well as wood density and climate niche.

## 2. Materials and Methods

### 2.1. Study Site

We sampled naturally occurring vegetation at three sites near Stanford, CA: Jasper Ridge Biological Preserve (37.40311, −122.24428), a 481-ha preserve in the eastern foothills of the Santa Cruz Mountains; the Stanford Quarry (37.39999, −122.16154); and the Stanford main campus (37.42875, −122.17919) (Figure 1). The climate is Mediterranean with a mean annual temperature of 14 °C, mean annual precipitation of 570 mm, and most rainfall occurring from November to April. The vegetation communities present at all three sites included chaparral and broadleaf, mixed evergreen forest. All three sites have loam to gravely clay loam soils derived from similar mixed sedimentary and metamorphic residuum and alluvium [31].

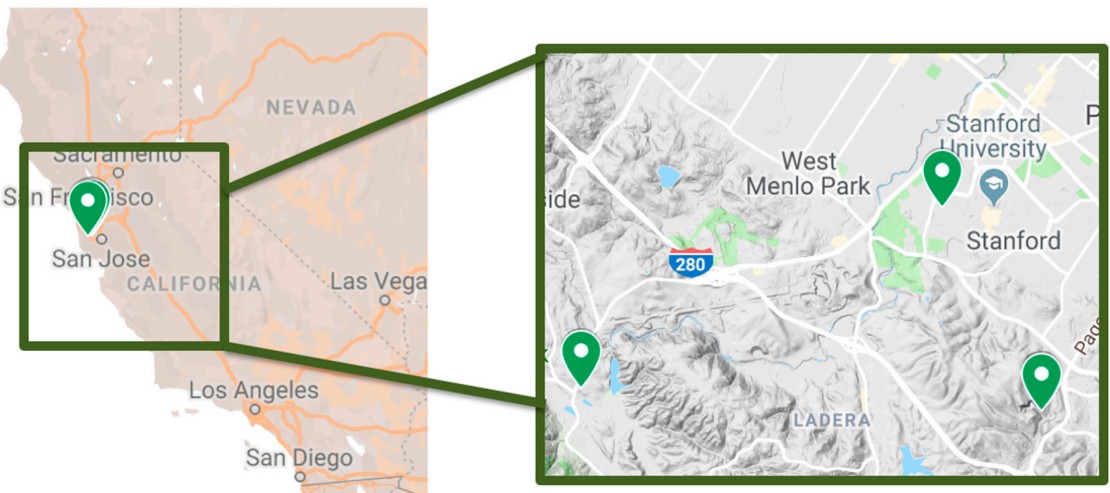

**Figure 1.** Map of study sites. Green markers represent the geographic location of each site.

## 2.2. Species Selection

We selected woody species native to northern California. In the spring and summer of 2017, we sampled five individuals from 13 native species growing on the Stanford campus or the Stanford Quarry. We sampled reasonably abundant species (five or more individuals in a ~200 m search area) and focused sampling on the largest individuals of each species present based on a visual survey. We included comparable data of wood density and plant allometry from Cornwell and Ackerly (2009) collected at the Jasper Ridge Biological Preserve, making for a total of 20 species of California trees and shrubs [19]. A range of habitats including riparian forests and chaparral/oak woodlands were sampled at all three sites to capture the dominant woody species across multiple microsites. We only sampled species that grew naturally at our sites. All three sites have similar climate and native plant communities. See Table S1 for a full list of species sampled and which datasets these species occurred in. For data on the heights, diameters, wood density values, and other traits measured of the study species, see Tables S2 and S3.

## 2.3. Trait Measurement

We obtained three branches per individual for at least five individuals sampled per species to use for measuring branch wood density via a volumetric method. We randomly selected mid- to upper-canopy, sun-exposed branches from each individual using hand pruners or a 4.5 m pole pruner. Branches of ~1 cm diameter and ~3 cm length were peeled of bark, their wet volume measured via mass displacement, and their dry weight measured following 72+ h drying at 65 °C. We measured individual height with an inclinometer. We obtained basal diameter for larger trees as diameter at breast height. We then calculated 'taper' as the basal diameter to height ratio, back calculating the basal diameter from diameter at breast height measurements assuming a conic stem where necessary. We averaged branch wood density per individual, and then averaged wood density and taper of individuals per species. We used branch wood density in order make our results comparable to wood density values obtained by Cornwell and Ackerly (2009) [19], who used data from three-year old branches from five individuals per species (see Preston et al. (2006) [18] for details of wood density measurements).

## 2.4. Habitat Preferences

We used information in the National Wildlife Federation and Audubon field guides as well as Calscape.org to determine whether species in our study were riparian specialists [32–34]. If the literature described a species as preferring rivers, streambanks, or riparian areas, we recorded these species as riparian specialists. Our dataset contained 11 riparian species and 9 non-riparian species with a mix of riparian and non-riparian species sampled at each of the three sites (Figure S1). Table S2 provides information on whether species were riparian and the sample size per species.

## 2.5. Paleohistory

We conducted a literature search in order to determine whether each species had ancestors in the Madro-Tertiary or Arcto-Tertiary geoflora. We assigned a geoflora category based off the most taxonomically specific level possible; for most species, we obtained geofloristic information at the genus level. For chaparral species, we placed them in the Madro-Tertiary group based on literature stating that the current chaparral community evolved from the Madro-Tertiary geoflora.

## 2.6. Climate Data

We determined climatic parameters describing each species climate niche using the WorldClim database and geo-referenced specimens from North America in the Global Biodiversity Information Facility (GBIF) database [35]. We used the full climate range for each species. We obtained 1960–1990 long-term averages of the following annual climatic parameters at 2.5 arc-minutes from the WorldClim

1.4 database [36]: Minimum temperature (Tmin), maximum temperature (Tmax), mean annual temperature (MAT), mean annual precipitation (MAP), and precipitation seasonality (coefficient of variation of precipitation across months, Pseason).We also obtained aridity data from the Global Aridity and PET Database [37], which is derived from WorldClim input data, including potential evapotranspiration (PET) and climate moisture deficit (CMD). CMD was calculated as the amount of water by which PET exceeds precipitation. Climate data were extracted for all GBIF specimen locations for each species. The extent of each species' climate niche was calculated based on the 5th and 95th percentile for each of the above climate variables.

*2.7. Data Analysis*

We averaged all wood density and taper values per species (taper did not show variation with plant height within-species) and constructed a series of linear models for the data, using R version 3.3.3. First, we constructed linear models relating wood density to taper, habitat preference, the interaction between the two, and the interaction without the habitat preference main effect. We selected the most parsimonious model based on the second-order Aikike information criterion (AICc, or AIC for small sample sizes) using the 'stats' package and the 'MuMIn' [38]. We also tested for univariate relationships between wood density and climate niche parameters related to water availability (CMD, MAP, PET) for the driest edge and wettest edge of each species (based on the 5th and 95th percentile of their species range), and then selected the best single climate variable by built linear models for niche parameters that were initially found to be significant using AICc. Finally, we combined the best structural model and the best climate model to build a single full model and all potential nested models, and then used AICc to select the most parsimonious of these models. The equation for our final model was as follows: WD ~5th% CMD + Riparian Specialization × Taper, where WD is wood density and CMD is climate moisture deficit. Finally, we verified that this inference was not purely driven by phylogenetic non-independence by refitting the final model using a phylogenetic generalized least squares (PGLS) model. We used the Smith and Brown 2018 [39] phylogeny (which contained all species except *Baccharis pilularis*, which was replaced with a congener to estimate branch lengths) and a Brownian motion covariance structure [40] using the *ape* [41], *geiger* [42,43], and *nlme* [44] R packages. See Figure S2 for the results of the phylogenetic analysis.

## 3. Results

A higher taper (basal diameter to height ratio) refers to woody plants that are wider in basal diameter at a given height (Figure 2). For riparian habitat specialist species, branch wood density decreased with increasing taper (Figure 2). For non-riparian species, no correlation existed between branch wood density and taper (Figure 2). The taper model with the lowest AICc had a main effect of taper and a taper x riparian interaction (but no riparian main effect), though this model was only a marginal improvement over a model with only a riparian specialization main effect ($\Delta$AICc = 1.1, likelihood ratio test $p = 0.06017$) (Table 1).

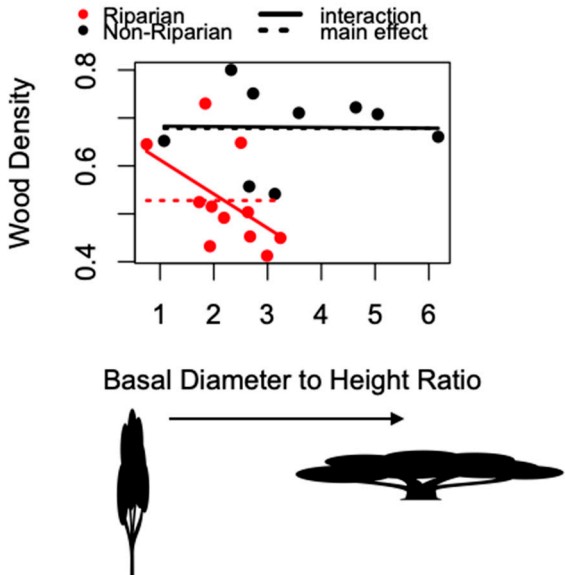

**Figure 2.** Branch wood density (g/cm$^3$) vs basal diameter to height ratio for California trees and shrubs. Trees with a low basal diameter to height ratio are tall and skinny, while trees with a high basal diameter to height ratio are short and wide, as represented by tree symbols. The arrow represents increasing basal diameter to height ratio. Red points/lines indicate riparian specialists and black points/lines non-riparian specialists. Dotted lines show the model fit for the habitat only linear model (Table 1), while solid lines show the model fit for the taper × habitat model.

**Table 1.** Model selection for structure and habitat specialization. WD refers to branch wood density. Taper is basal diameter to height ratio. Habitat refers to whether or not species are riparian specialists. AIC, Aikike information criterion.

| Model | Degrees of Freedom | AICc |
|---|---|---|
| Null (WD~1) | 2 | −24.22 |
| WD~taper | 3 | −22.29 |
| WD~habitat | 3 | −31.84 |
| WD~taper + habitat | 4 | −29.22 |
| WD~taper + habitat + taper/habitat | 5 | −29.49 |
| WD~taper + taper/habitat | 4 | −32.95 |

Few correlations existed between wood density and species driest range edge, on the contrary to the existing relationships between wood density and site water availability in global meta-analyses (Figure 3). However, positive relationships between wood density and the dryness of a species' wettest range boundary were found for all metrics of moisture availability (5th percentile CMD, PET, and MAT; 95th percentile MAP), with 5th percentile CMD being the best climate predictor of wood density (Table 2). Riparian specialists had generally lower wood density than non-specialists for the same wet edge aridity, but showed a similar positive relationship between wood density and wet edge aridity (Figure 3). Adding a main effect of habitat specialization improved the best climate model (ΔAICc = 6.86, Table 2), but adding an interaction between climate and habitat specialization did not improve the model. These results were qualitatively similar if climate niches were constructed from only California occurrence records in the California Consortium of Herbaria [45].

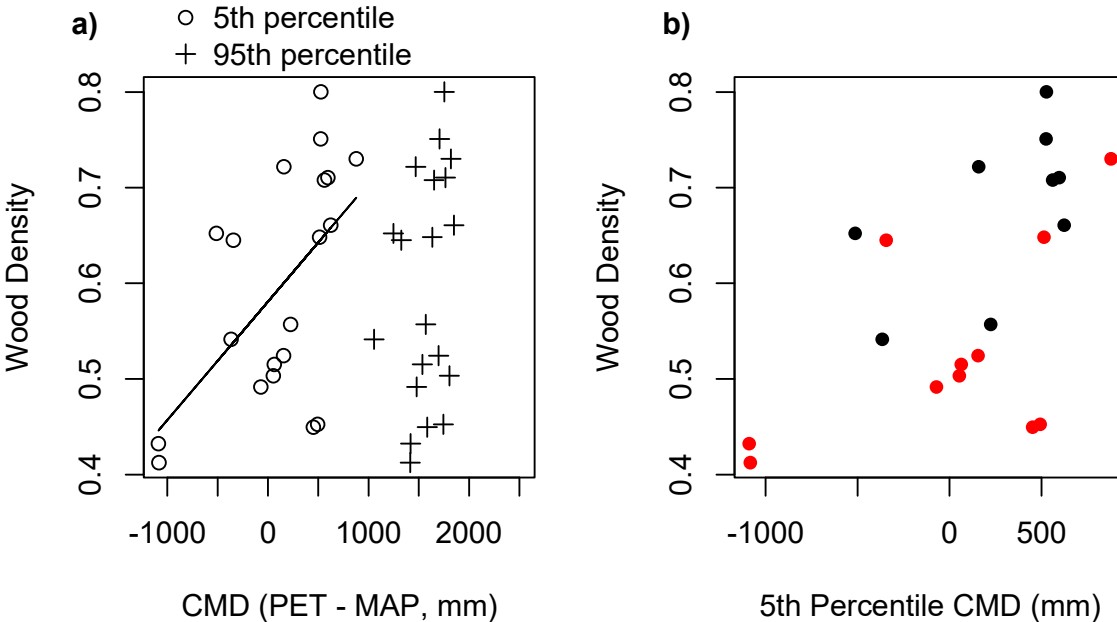

**Figure 3.** Branch wood density vs. (**a**) species 5% and 95% climate moisture deficit (CMD) and (**b**) species 5% CMD for riparian vs. non-riparian California trees and shrubs. Black dots represent non-riparian specialist species, while red dots represent species that are riparian habitat specialists. The black line indicates a significant relationship between wood density and CMD at the 5th percentile, but no significant effect at the 95th percentile of CMD. PET, potential evapotranspiration; MAP, mean annual precipitation.

**Table 2.** Model selection for wet edge climate effects. WD refers to branch wood density. Habitat refers to whether or not species are riparian specialists. CMD 0.05 is the 5th percentile climate moisture deficit. MAP 0.05 is the 5th percentile for mean annual precipitation. MAT 0.05 is the 5th percentile for mean annual temperature. PET 0.05 is the 5th percentile for potential evapotranspiration.

| Model | Degrees of Freedom | AICc |
|:---:|:---:|:---:|
| Null | 2 | −21.20 |
| WD~CMD 0.05 | 3 | −29.21 |
| WD~MAP 0.05 | 3 | −22.04 |
| WD~PET 0.05 | 3 | −28.84 |
| WD~MAT 0.05 | 3 | −25.75 |
| WD~CMD 0.05 + habitat | 4 | −36.07 |
| WD~CMD 0.05 + habitat/CMD 0.05 | 4 | −27.80 |
| WD~CMD 0.05 + habitat + habitat/CMD 0.05 | 5 | −32.58 |

Although geoflora suggested small differences between the climate niches of Madro-Tertiary and Arcto-Tertiary species, especially for the minimum temperature range (Figures 4 and 5), geoflora did not directly affect wood density ($\Delta$AICc = 0.62918, likelihood ratio test $p$ = 0.2794) or strongly differentiate the climate niches of modern California plant communities (Figures 4 and 5).

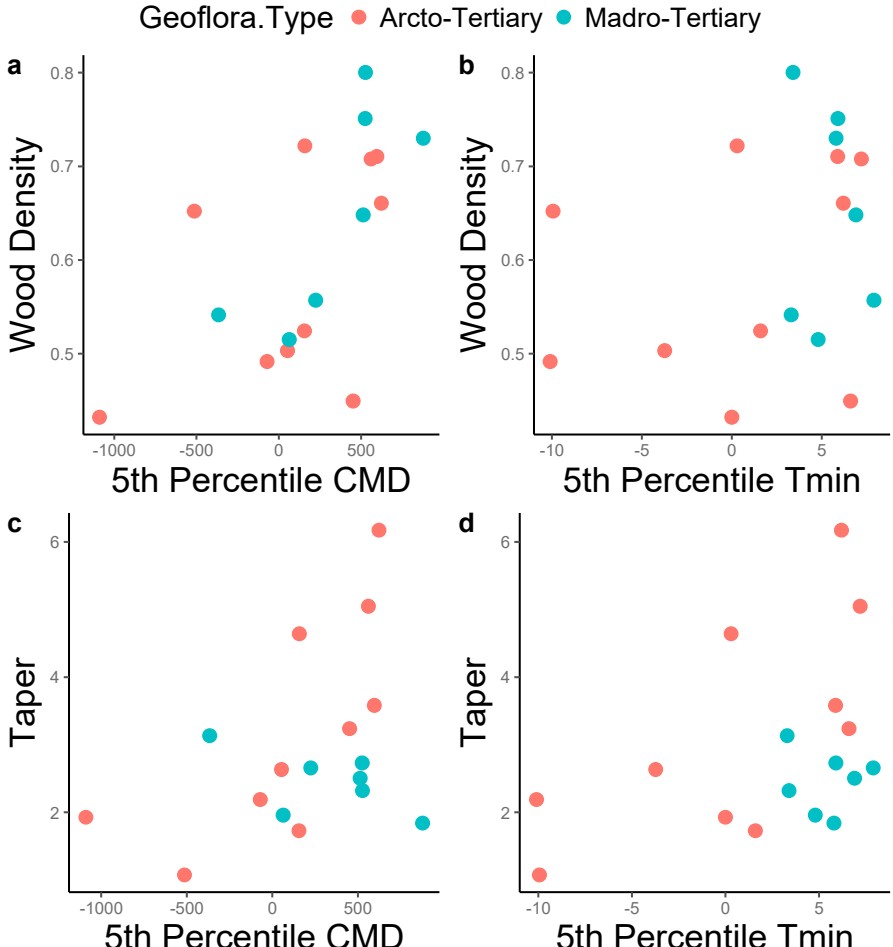

**Figure 4.** Branch wood density g/cm³ vs. (**a**) 5th percentile climate moisture deficit (mm) and (**b**) 5th percentile Tmin (°C) for California trees and shrubs. Taper vs. (**c**) mean climate moisture deficit and (**d**) 5th percentile Tmin for California trees and shrubs.

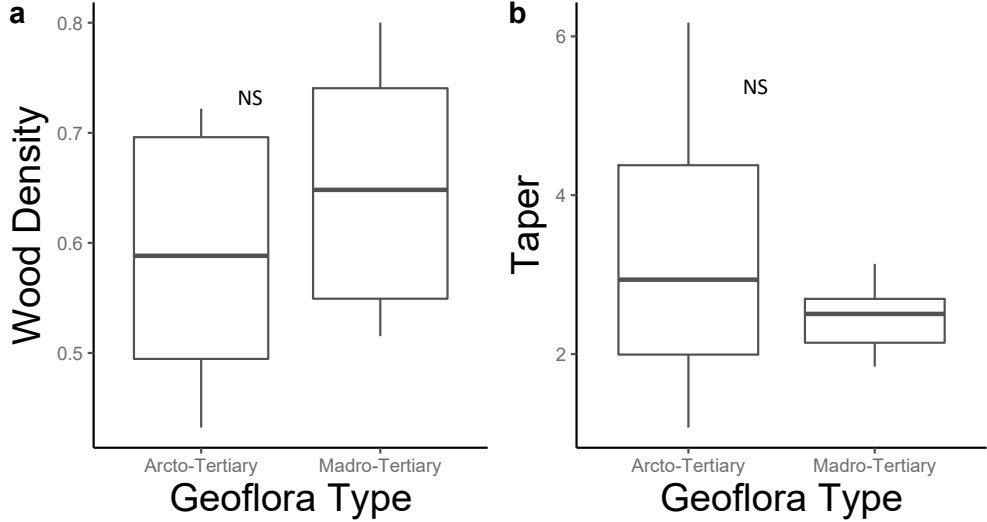

**Figure 5.** (**a**) Boxplot of wood density values and (**b**) taper (basal diameter to height ratio) values per geoflora type. NS denotes non-significance.

Combining all evidence, aridity of the wet range edge and, for riparian specialists, taper explained almost three quarters of the variation in wood density across this California plant community.

The most parsimonious full model overall included a main effect of 5th percentile CMD and a riparian specialization x taper interaction (Table 3, Figure 6) and had an $R^2$ of 0.74. On the basis of a decomposition of variance analysis (assessing the $R^2$ of nested climate and structural models), structure and habitat specialization explained more of the variance than climate, with species 5th percentile CMD explaining 32% of the total variance and the taper × riparian specialization interaction explaining 52% of the variance, with 10% of the variance shared between them. All model effects remained significant and qualitatively similar when controlling for phylogenetic non-independence using a PGLS model [39], though visual inspection of the congeneric species in the dataset suggested that the lack of relationship between taper and wood density for non-riparian species may be driven by the high wood density and high taper values of the four species of oaks in the dataset (Figure S2).

**Table 3.** Model selection for most parsimonious model overall. WD refers to branch wood density. Taper is basal diameter to height ratio. Habitat refers to whether or not species are riparian specialists. CMD 0.05 is the 5th percentile climate moisture deficit.

| Model | Degrees of Freedom | AICc |
|:---:|:---:|:---:|
| Null (WD~1) | 2 | −21.20 |
| WD~CMD 0.05 | 3 | −29.21 |
| WD~CMD 0.05 + taper + habitat + taper/habitat | 6 | −37.76 |
| WD~CMD 0.05 + taper + taper/habitat | 5 | −41.74 |
| WD~taper + habitat + taper/habitat | 5 | −29.49 |
| WD~CMD 0.05 + habitat | 4 | −36.07 |
| WD~habitat | 3 | −31.84 |
| WD~taper | 3 | −22.29 |

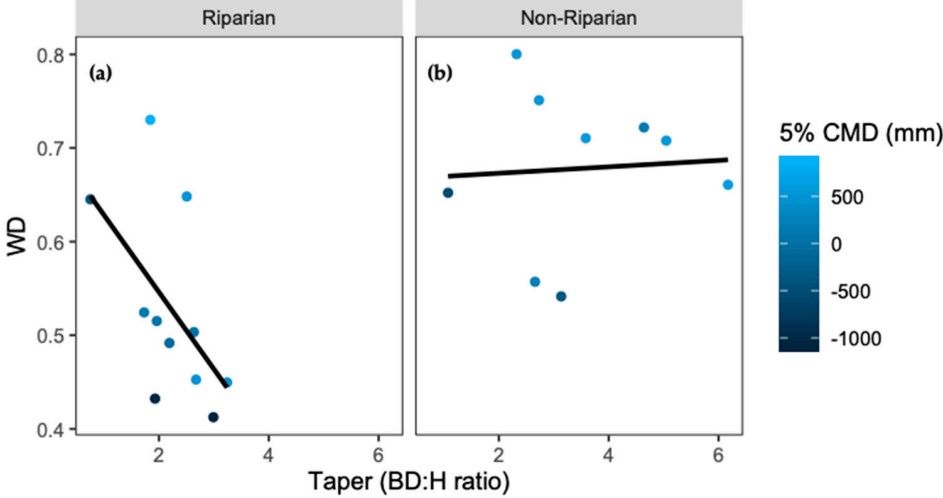

**Figure 6.** Branch wood density vs basal diameter to height ratio (taper) for (**a**) riparian and (**b**) non-riparian species, colored by the 5th percentile climate moisture deficit shown for each species (drier in lighter blue). Trend lines are only for the wood density (WD)~taper relationships.

## 4. Discussion

In a Mediterranean woody plant community, we found that canopy allometry was only related to wood density in riparian specialists, that geofloristic provenance had little influence on wood density, and that species climate niche had unexpected relationships with wood density. These results contrast with both global analyses of wood density and site-specific investigations of wood density variation within individual communities. We discuss the implications of our results for the context dependence

of the climatic and structural controls on wood density, and the potential for disconnects between small-scale environmental filtering at microsites and large-scale environmental filtering of a regional species pool.

We found that allometry only relates to wood density in riparian contexts. In the California mixed evergreen forest, the relationship between wood density and allometry, however, does not appear as consistent as the wood density–allometry structural equations proposed by Chave et al. (2005) for tropical forests [13]. This lack of consistency may be explained by the differing functional strategies of riparian and non-riparian species. We found that riparian specialists had a negative relationship between wood density and taper consistent with Chave et al. (2005) [13], indicating that species with high wood density are taller at a given diameter than species with low wood density. This result is also consistent with Iida et al. (2012), who found that species with higher wood density have smaller diameters at a given height in Malaysia [16]. However, these findings contrast with those of Aiba & Nakashizuka (2009) as well as King, Davies, Tan, & Noor (2006), who found no relationship between wood density and height for a given diameter in tropical forests [14,15]. Predictions from engineering theory suggest that high wood density does not necessarily increase the mechanical stability of taller trees at a given diameter, which further contrasts with our results [46]. This negative relationship between wood density and taper is consistent with previous work at one of our field sites that showed strong relationships between wood density, allometry, and microsite soil moisture [18,19]. This allometric relationship with wood density for riparian plants may be similar to allometry–wood density relationships in the wet tropics, because both ecosystems are not water-limited.

Differences in allometry–wood density relationships between riparian and non-riparian species may reflect evolutionary processes. Chave et al. (2009) suggested that wood density is related to water transport efficiency [1]. Stronger selection on wood density–allometry relationships may, therefore, occur in wet climates where efficient transportation of water would be advantageous. Non-riparian species, however, lack this wood density–allometry relationship, suggesting that, in water-limited ecosystems, these species are 'overbuilt' for other reasons [47,48]. Riparian forests tend to be more light-limited than immediately adjacent non-riparian forests owing to their closed, tall canopies, so selection may occur in riparian habitats to favor taller, competitive trees, giving wood density more of a structural role in riparian than non-riparian ecosystems, but not in systems with less light competition. The high taper, non-riparian species were all oaks, including species with deep roots that can penetrate groundwater [49,50]. A large dataset covering more sites would be needed to determine if the oaks have a phylogenetic signal.

The California floristic province formed from the confluence of a diverse range of geofloras [29], but this history did not explain the variation in wood density. The Madro-Tertiary geoflora has been associated with the modern chaparral species in California and associated with the development of a Mediterranean climate in California [30]. Species with Madro-Tertiary ancestry did not significantly differ in climate niche than species with temperate Arcto-Tertiary history, suggesting that paleoclimatic influences do not clearly shape modern climate niches in this community. There were neither strong influences of paleoclimatic history on wood density directly nor on the effects of climate or taper on wood density. A more comprehensive study of the traits and climate niches of Madro-Tertiary versus Arcto-Tertiary geofloras could reveal more subtle legacies of their biogeographic origins, but we found little preliminary evidence of geofloristic legacies in this community.

Surprisingly, the wet edge of climate moisture deficit best predicted wood density when compared with other climate variables. This contrasts with meta-analyses conducted at regional and global scales where mean annual temperature and mean annual precipitation of the sampling site explained patterns in wood density across many species [11,20,21] and higher wood density was associated with decreased drought mortality [26]. The positive relationship between wood density and climate moisture deficit is consistent with previous literature, but it is surprising that this trend does not appear for the dry edge. For a Mediterranean plant community, Camarero (2019) found that the relationship between April precipitation and tree growth was the strongest in the species with the highest wood density and that

the wet season particularly influenced radial growth rates [6]. In contrast, for a Mediterranean climate in South Africa, xylem density positively correlated with seasonal water stress, suggesting that the dry edge of climate niche was more of an influence [10]. For tropical and subtropical forests, wood density positively related to aridity across sites [22,25,27]. Meanwhile, in a tropical forest, deciduous versus evergreen leaf habit mediated the degree of variation in wood density in response to rainfall [51]. A water-stressed Mediterranean climate may select for lower wood density in wet environments rather than higher wood density in dry environments. Structural and competitive constraints may have more of an influence on wood density in wetter ecosystems, where it is more advantageous to grow tall and fast. Selection for extremely efficient water transport may occur in wetter climates where increased conductivity has an advantageous impact on plant fitness.

In order to better understand landscape-level variation in wood density, a more nuanced approach is required beyond the species-mean climate–wood density correlations used in global analyses. Moreover, examining relationships between climate and wood density alone may overlook the influence of structural constraints and habitat specialization on wood density. Structure influenced wood density only for riparian species, suggesting that the relationship between wood density and allometry is highly context-specific, and that differences in habitat specialization across a landscape should be accounted for. A disconnect often exists between the climate and wood density correlations observed at regional levels and the microsite influences on wood density at localized scales [6,11,18–20]. Our findings caution against the overgeneralization of how structural constraints influence wood density across ecosystems. Further research is needed to explore the complexity of factors, such as climate, allometry, and habitat specialization, that shape wood density variation at a community level. Understanding how these factors drive functional tradeoffs in wood traits can provide more nuanced insights into the biological mechanisms by which these tradeoffs occur. Elucidating the extent to which climate and local habitat explain wood functional traits is relevant in the context of increasing water stress resulting from anthropogenic climate change [26,52,53]. Furthermore, fire greatly shapes the structure and composition of California ecosystems and relates to wood density in tropical forests, yet the extent to which wood density relates to species' responses to fire remains unknown [54]. Finally, an exploration of within-species patterns of wood density and allometry between individuals might elucidate some of the fundamental structural constraints driving the patterns we found in riparian specialists in this comparative study.

## 5. Conclusions

In summary, we found that a negative relationship occurred between the basal diameter to height ratio and wood density only for riparian habitat specialists. Geofloristic history had little influence on relationships between wood density and climate in our woody plant community. The wet edge of climate moisture deficit further explained variation in wood density. These results suggest a more nuanced approach is required in order disentangle the drivers of wood density variation at a community level.

**Supplementary Materials:** The following are available online at http://www.mdpi.com/1999-4907/11/1/105/s1, Tables S1–S3 and Figures S1 and S2.

**Author Contributions:** Conceptualization, R.A.N., E.J.F., J.A.B., W.K.C., and L.D.L.A.; methodology and sampling design and data collection, R.A.N. and L.D.L.A.; analysis, R.A.N. and L.D.L.A.; data curation, R.A.N., W.K.C., and L.D.L.A.; writing—Original draft preparation, R.A.N., E.J.F., and L.D.L.A.; writing—Review and editing, R.A.N., E.J.F., J.A.B., W.K.C., and L.D.L.A.; visualization, R.A.N. and L.D.L.A.; supervision, L.D.L.A.; project administration, R.A.N. and L.D.L.A.; funding acquisition, R.A.N. and L.D.L.A. All authors have read and agreed to the published version of the manuscript.

**Funding:** This research was funded by a Stanford Vice Provost for Undergraduate Education small grant. L.D.L.A. received funding from the National Science Foundation (DBI-1711243) and a Climate and Global Change postdoctoral fellowship from the National Oceanic and Atmospheric Administration. Any opinions, findings, and conclusions or recommendations expressed in this material are those of the author(s) and do not necessarily reflect the views of the National Science Foundation.

**Acknowledgments:** Thank you to Rodolfo Dirzo for providing initial advice on this project.

**Conflicts of Interest:** The authors declare no conflict of interest. The funders had no role in the design of the study; in the collection, analyses, or interpretation of data; in the writing of the manuscript; or in the decision to publish the results.

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
