# Peer review of "The Role of Climate Niche, Geofloristic History, Habitat Preference, and Allometry on Wood Density within a California Plant Community"

_forests, doi:10.3390/f11010105_

Round 1
Reviewer 1 Report
The work presented for the review concerns the role of climate niche, geofloristic history, habitat preference and allometry on wood density within California plan community. Overall, I found that the work is interesting, the analysis is well performed and study set-up ok. However, I have same comments for the authors. The section “Material i methods” needs expansion in some areas and clarification in others.
Major comments on text:
Material & methods
In my opinion additional information about woody species, climatic data and data analysis is needed.
2.2 Species Selection
some information about species, I have a question: what species were studied? some information about diameter and height of selected trees and shrubs.2.3 Trait Measurement
there is no information how the branches were obtained. I think there should be a notice about the height/ place of branch sampling and the information if it was always the same. If the sampling method is not important, this information should also be in this section.2.4 Habitat Preference
- there is no information about the number of riparian and non-riparian species used in the study, that’s why I have the questions:
How many riparian species/samples and how many non-riparian species/samples were used in the study? Was the number of samples from the riparian species and other species was comparable?2.5 Climate Data
There is no information of the time range of climatic data used in study. Additionally, the authors didn’t specify whether they have used annual or seasonal PET values. In lines 147-148 authors mentioned both annual and seasonal precipitation.
2.6 Data Analysis
There should be also added a little more information about the final model.
Minor comments on text:
Lines 110-112: “The climate is Mediterranean with (mean annual temperature: 14°C, mean annual precipitation: 570mm) with rainfall occurring from November to April”. - that sentence (brackets) should be corrected.
Line 155: “We averaged all WD and taper values per species……..” - the abbreviation “WD” should be explained here
Figure 3a: the title of the vertical axis - should be wood density instead of the abbreviation “WD”
Reviewer 2 Report
Dear authors,
first would like to thank you presenting of your results.
I have few question and suggestions:
103) Hypothesis No.4 - The clarity of formulation of this hypothesis e.g. climatic relationship with climate; high spatial heterogenity- is not described in following text
107) Are the sites geobotanicaly identical?
116) Species selection - You sampled all of 13 species in all of study areas? If not (in case that the sites differ in microsites characteristics) can unequal sample size together with site effect strongly affect the results.
129) The average of wood density per species (maybe describe if exist some variability between individuals) could affect the results? Why you used the density of branches?
154) Data analysis - Would be nice to present the basic equation of your model/s.
Fig.3 - Unify in the legend WD or Wood density
Fig.4. - Legibility of this picture is low (special the legend of axis and marks). Maybe devide in two pictures
Fig.5. - Which tren lines are presented? CMD in to description of picture (other pictures have it).
228) Discusion - Do you tested also sites separately? Could it explain microsite effect mentioned ind discusion?
237) "constrain" better formulation?
273) Could be possible obtain better corelations in case of using late and early wood density analysis?
300) Some literature or its your finding?
314) Maybe to general claim ("Mediterranean"). "In our study sites/communities"?
Best Regards
Reviewer 3 Report
The introduction L43 mentions that phylogeny might influence these relationships, and several species in the study are con-generic. Although including phylogenetic information in the analysis may not change the conclusions of this study, it seems like a missed opportunity to address the potential for similarity among close-relatives. If a phylogenetically informed analysis was included, or showing figures with con-generic species in the same color, this would provide confidence in the results for readers that may be concerned about the effect of phylogenetic non-independence.
L55 missing word? do you mean that higher wood density is associated with smaller tree diameters?
L65 not clear what is meant by “can influence variation in tree allometry”
L87 I think many ecologists will be unfamiliar with the term geofloristics. I like this definition for geoflora, I think it could come earlier at the first mention of the term geofloristic. Also, there could be a better explanation of Madro- and Arcto-Tertiary are these separated in space or in time? not quite clear what the distinction means. Also on L102-103 why is it predicted that Madro- would have stronger wood density-climate relationships than Arcto-?
L90 Acrto -> Arcto
L109 are the Stanford Quarry and Stanford main campus natural areas, or are these planted areas? IF planted, there needs to be more information about conditions
Figure 3 y-axis says WD -> Wood density
Figure 4 Symbols, axis text and legend in A-D need to be larger. For E the axis text needs to be larger
Figure 5 hard to see differences in colors of dots, suggest to define colors so that differences in the scale can be seen
L245-249 this sentence is very long to follow, suggest to separate out and more completely explain the last clause as another sentence.
L267 extra “in”
L290 “where wood density is related to conductivity” seems odd, because wood density will be related to conductivity in all habitats. Do you mean that higher wood density might be adaptive in riparian areas where increases in conductivity could have a substantial impact on plant fitness, while in drier areas resistance to embolism becomes more important to fitness?
Also L290 this snow loading hypothesis needs to be put into the context of the sites examined for readers not familiar with these ecosystems, were exactly would the snow be located? It would be helpful to have a map that showed the climatic zones that were defined and how they relate to the three sites and the species distributions.
